# Putative Biomarkers for Acute Pulmonary Embolism in Exhaled Breath Condensate

**DOI:** 10.3390/jcm10215165

**Published:** 2021-11-04

**Authors:** Inger Lise Gade, Jacob Gammelgaard Schultz, Rasmus Froberg Brøndum, Benedict Kjærgaard, Jens Erik Nielsen-Kudsk, Asger Andersen, Søren Risom Kristensen, Bent Honoré

**Affiliations:** 1Department of Hematology and Clinical Cancer Research Center, Aalborg University Hospital, 9000 Aalborg, Denmark; rfb@rn.dk; 2Department of Clinical Medicine, Aalborg University, 9000 Aalborg, Denmark; srk@rn.dk (S.R.K.); bh@biomed.au.dk (B.H.); 3Department of Clinical Biochemistry, Aalborg University Hospital, 9000 Aalborg, Denmark; 4Department of Cardiology, Aarhus University Hospital, 8200 Aarhus, Denmark; jacobgschultz@clin.au.dk (J.G.S.); nielsen-kudsk@clin.au.dk (J.E.N.-K.); asger.andersen@clin.au.dk (A.A.); 5Department of Clinical Medicine, Faculty of Health, Aarhus University, 8200 Aarhus, Denmark; 6Department of Cardiothoracic Surgery, Aalborg University Hospital, 9000 Aalborg, Denmark; benedict.kjaergaard@rn.dk; 7Department of Biomedicine, Aarhus University, 8000 Aarhus, Denmark

**Keywords:** pulmonary embolism, breath test, proteomics, diagnosis, animal models, mass spectrometry

## Abstract

Current diagnostic markers for pulmonary embolism (PE) are unspecific. We investigated the proteome of the exhaled breath condensate (EBC) in a porcine model of acute PE in order to identify putative diagnostic markers for PE. EBC was collected at baseline and after the induction of autologous intermediate-risk PE in 14 pigs, plus four negative control pigs. The protein profiles of the EBC were analyzed using label-free quantitative nano liquid chromatography–tandem mass spectrometry. A total of 897 proteins were identified in the EBCs from the pigs. Alterations were found in the levels of 145 different proteins after PE compared with the baseline and negative controls: albumin was among the most upregulated proteins, with 14-fold higher levels 2.5 h after PE (*p*-value: 0.02). The levels of 49 other proteins were between 1.3- and 17.1-fold higher after PE. The levels of 95 proteins were lower after PE. Neutrophil gelatinase-associated lipocalin (fold change 0.3, *p*-value < 0.01) was among the most reduced proteins 2.5 h after PE. A prediction model based on penalized regression identified five proteins including albumin and neutrophil gelatinase-associated lipocalin. The model was capable of discriminating baseline samples from EBC samples collected 2.5 h after PE correctly in 22 out of 27 samples. In conclusion, the EBC from pigs with acute PE contained several putative diagnostic markers of PE.

## 1. Introduction

The diagnostic workup for pulmonary embolism (PE) includes clinical examination evaluating the clinical probability of PE, D-dimer testing, and imaging diagnostics [1]. However, neither the clinical presentation nor the initial tests are specific for PE [2,3]. Therefore, there is an unmet need for diagnostic biomarkers to enhance the early and precise diagnosis of PE.

The air we exhale contains water vapor, volatile organic compounds and low concentrations of proteins; however, these concentrations are still high enough for mass spectrometry analysis. This biological specimen, called exhaled breath condensate (EBC), can be collected non-invasively in both conscious and intubated individuals [4]. More than a hundred proteins have been identified in the EBC [5,6,7,8]. In a recent pilot study based on seven pigs with PE, we identified 131 proteins in the EBC [9], and 79 of these proteins were also identified by mass spectrometry in a human study using the same EBC collection device [10]. Along with the similarity of the study objects in animal studies, the porcine model has considerable advantages in the search for biomarkers, since it enables us to strictly follow the course of PE from infusion of a thrombus into the lungs. 

The objective of this study was to search for putative diagnostic markers for PE in the EBC using the porcine model. EBCs were collected from pigs before and after the induction of autologous intermediate-risk PE and from negative controls without PE.

## 2. Materials and Methods

### 2.1. Study Design

In total, six EBC samples were collected from each pig; two samples before the induction of PE (pooled into one Pre PE sample), two samples 30 min after the induction of PE (pooled into one Early Post PE sample), and two samples 2.5 h after PE (pooled into one Late Post PE sample) and correspondingly in the negative controls (Pre C, Early Post C and Late Post C) (Figure 1). The two EBC samples from each time point were pooled into 2 mL tubes and stored at −80 °C until being analyzed as one sample with regard to protein concentration and in the subsequent mass spectrometry analysis.

This study was an amendment protocol to a pharmacological study on the effects of vasodilatory drugs in a porcine model of intermediate-risk PE [11]. Four increasing equivalent doses of vasodilatory drugs (either nitric oxide [NO], Sildenafil, or Riociguate) were administered to the pigs in the pharmacological study. The first two (very low and low) doses of vasodilatory drugs were given before the collection of the Late Post PE sample in eight of the pigs (Figure 2). In order to avoid the possible influence of the vasodilatory drugs on the EBC samples, the Late Post PE EBC samples were collected immediately after administration of the second dose of the study drugs. 

We compared the protein profiles of both Early Post PE and Late Post PE samples with Pre PE samples and the corresponding negative controls, as shown in Figure 1, in order to detect most of the putative markers of PE. The proteomes of the individual EBC samples were analyzed using quantitative label-free nano liquid chromatography–tandem mass spectrometry (LFQ nLC-MS/MS). The proteome data were analyzed in two steps. First, in the discovery phase, differentially expressed proteins in EBC samples were identified by a simple students t-test without correction for multiple tests in order to detect all potential markers of PE. Second, prediction models and differential expression analysis including correction for multiple comparisons qualified the putative markers of PE in the EBC. 

### 2.2. Research Animals

The study animals were female Danish Landrace pigs weighing 60 kg. Negative control pigs underwent the exact same protocol regarding medication, intubation, monitoring, ventilation, placement of sheaths, etc., as the pigs with induced PE. The negative controls were given a bolus of isotonic saline in the step in which the PE was infused in the PE pigs (Figure 1). 

EBC was collected from 22 pigs in total; four negative controls and 18 pigs with autologous PE (Figure 2). The Danish Animal Experiment Inspectorate approved the study protocol (license number 2016-15-0201-00840), and national and international guidelines concerning the ethical care of experimental animals and Danish legislation on transport of livestock were followed. 

The study protocol concerning anesthesia, ventilation, monitoring and the induction of pulmonary embolism in the pigs was previously thoroughly described [11]. Briefly, anesthesia was induced by means of intravenous etomidate (0.5 mg/kg, Hypnomidate^®^ Janssen Pharmaceutical, Beerse, Belgium) and maintained after intubation (ID 7.5 mm, Unomedical, Sungai Petani, Malaysia) by means of the continuous intravenous infusion of propofol (2 mg/kg/hour, Propolipid, Fresenius Kabi, Bad Homburg, Germany) and fentanyl (5 µg/kg/hour, Hamlen Pharma, Hamlen, Germany). Upon intubation, pressure-controlled volume-gated mechanical ventilation (GE Datex-Ohmeda S/5 Avance) with non-humidified air was initiated. To correct hypoxemia in association with intubation, the fraction of inspired O_2_ was initially set to 1.0, but reduced to 0.3 soon after. The tidal volume was set to 8 mL/kg and the respiratory rate to 16 breaths per minute, no positive end-expiratory pressure was applied. The end tidal CO_2_ before the induction of the PE was maintained at 5.0 kPa by increasing the respiratory rate if necessary.

The core temperature was maintained at 38–39 °C using the Bair Hugger^TM^ normothermia system, monitored continuously with a rectal thermometer. Three intravascular sheaths (6–7 French) were inserted guided by ultrasound before the first collection of EBC for continuous isotonic saline infusion and drawing of venous blood samples, continuous monitoring of blood pressure and heart rate and arterial blood sampling, and infusion of anesthetics and invasive hemodynamic measures. The right ventricle end systolic volumes and the left ventricle end systolic volumes were measured by means of transoesophageal echocardiography.

Sixty milliliters of blood was drawn from the sheath in the left external jugular vein after the first collection of EBC. The venous blood was distributed in two 4.75” uncoated extra corporal tubes for spontaneous coagulation forming two autologous emboli at room temperature over the next three hours. A 26-French Dry-Seal sheath (Gore Medical, Newark, DE, USA) was placed via the right external jugular vein at the superior vena cava ending at the level of the right atrium for the administration of the emboli and right-sided heart catheterization before and after the induction of PE. Serum was discarded before each embolus was transferred to the 26-French sheath, through which it was flushed by isotonic saline into the right atrium to travel with the blood flow and lodge in the pulmonary arteries. The pigs were euthanized by means of an intravenous injection of phenobarbital (67 mg/kg, Exagon^®^, vet, Richter Pharma, Wels, Austria) at the end of the experiment.

### 2.3. Collection and Storage of the Exhaled Breath Condensate 

The exhaled air was cooled, and the resulting condensate collected in a polypropylene collection tube precooled to −80 °C (RTubeVent, Respiratory Research Inc., Austin, TX, USA). The collection tube and the associated cooling sleeve was inserted between the expiratory limb of the Y-connector and the expiratory limb of the mechanical ventilator (Figure 3). The heat and moisture exchange filter was disconnected during collection of the EBC. Gowns, surgical hairnets and new nitrile gloves were worn whenever the EBC was handled. The EBC was transferred to 2.0 mL low-temperature freezer vials and stored at −80 °C immediately after collection.

A condensate sample from the mechanical ventilator was collected by inserting a sterile set of ventilator hoses connected to an adult breathing circuit with the Evaqua™ technology (RT380/RT385, Fisher & Paykel Healthcare, Auckland, New Zealand). The moisture chamber was filled with sterile, isotonic sodium chloride. The resulting condensate was collected by insertion of the collection tube covered by the aluminum sleeve precooled to −80 °C inline at the expiratory limb directly at the mechanical ventilator. 

### 2.4. Preparation of EBC Samples for Mass Spectrometry

The volume of EBC was estimated by weighing, assuming that 1 g equals 1 mL. Samples were vacuum-centrifuged and re-dissolved in 120 µL of digestion buffer (0.5% SDC, 20 mm tetraethylammonium bromide). The protein concentration was estimated by measuring the fluorescence (excitation at 295 nm, emission at 350 nm) under the presumption that 1 g of protein corresponds to 0.0117 g of tryptophan, as is the case for human and mouse protein samples [12]. A standard curve was constructed from tryptophan. 

Samples were incubated with tris(2-carboxyethyl) phosphine at 55 °C for 1 h. Iodoacetamide was added and samples were incubated for a further 30 min at room temperature protected from light, before being added to Microcon 30 K centrifugal filter devices (Merck Milipore Ltd., Tullagreen, IRL, Cork, Ireland) and centrifuged at 14,000× *g* for 15 min. Digestion buffer was added to the filter units twice with centrifugation. Finally, 50 µL of digestion buffer was added together with trypsin and mixed at 600 rpm in a thermomixer for 1 min. The units were incubated in a wet chamber at 37 °C overnight. Samples were then centrifuged for 14,000× *g* for 10 min into new collection tubes. The filters were rinsed with 100 µL digestion buffer twice. Trifluoroacetic acid was added to a final concentration of 0.5% (*v*/*v*). An equal volume of ethyl acetate was added, shaken for 1 min and centrifuged at maximum speed for 2 min. The lower phase was saved by the removal of the upper phase. The extraction was repeated two times. The samples were dried in a vacuum centrifuge and re-suspended in 120 µL of 100 mM tetraethylammonium bromide. The peptide concentration was measured by fluorescence. Samples were vacuum dried, re-suspended in 0.1% (*v*/*v*) formic acid at a concentration of 0.05 µg/µL. Three microliters (0.15 µg) were loaded for each run and samples were analyzed in replicates. Some samples caused problems during the LC run, mostly because of pressure increases. Files from five of the EBC collections were therefore not replicated.

### 2.5. Nano Liquid Chromatography–Tandem Mass Spectrometry (nLC-MS/MS)

This was performed with the following modifications as previously described [9]. An analytical column (EASY-Spray Column, 750 mm × 75 µm, PepMap RSCL, C18, 2 mm, 100 Å, Thermo Scientific, Boston, MA, USA) was used to separate peptides. A 90 min gradient was formed by mixing buffer A (0.1% formic acid) with buffer B (80% acetonitrile, 20% water, 0.1% formic acid). The following amount of buffer B was used: 6% (0 min), 16% (3 min), 30% (55 min), 60% (67 min), 99% (70 min), 99% (80 min), 6% (81 min), 6% (90 min).

### 2.6. Identification and Label-Free Quantification of the Proteins 

The raw MS datafiles were searched against Uniprot databases, Sus scrofa and Homo sapiens downloaded on February 22 2017 using MaxQuant (v1.5.5.1) [13] for label-free quantification (LFQ) analysis, as previously described [9]. The false discovery rate (FDR) was set to 1% at the peptide spectrum match level, as well as at the protein level. 

### 2.7. Differential Expression Analysis for Discovery of Putative Markers

The protein expression data from MaxQuant were entered into Perseus (v1.5.8.5) [14] for data management and discovery differential expression analysis. Proteins identified as potential contaminants, only identified by site or by the reverse part of the database, were removed. The proteins included in the subsequent analyses are not necessarily represented in all the EBC samples, since we chose not to filter by valid values in this first step of the data analysis (i.e., we did not remove proteins if they were not found in all the EBC samples and we accepted proteins identified with one peptide because we were interested in all possible proteins in the samples in this explorative study). All peptides were used for the protein quantification. The LFQ values were log_2_-transformed, and the arithmetic means of the technical duplicates were calculated. Protein levels before and after PE were compared using unpaired two-sided *t*-tests, and differences were calculated as fold changes of the LFQ values (e.g., amount of a certain protein in Late Post PE samples divided by the amount of the same protein in Pre PE samples). Linear fold changes were calculated by taking the base 2 to the power of the log_2_-transformed fold changes. Fold changes > 1 indicate that the protein had higher LFQ values in the particular type of sample (i.e., pigs compared with the mechanical ventilator; in the PE animals as compared with the negative controls; in the Early Post PE or Late Post PE as compared with the Pre PE in the PE pigs; and Early Post C or Late Post C compared with the Pre C in the negative controls, respectively). Fold changes < 1 indicate that the proteins had lower LFQ values in the particular type of sample in the same comparisons. A linear fold change of 17.1 is referred to as a 17-fold higher amount/upregulation of the protein, while a fold change of 0.16 corresponds to a protein amount of 1/6. Proteins with *p*-values < 0.10 were considered significantly changed after PE, chosen at this level in order not to miss possible candidates (to avoid type 2 faults because of the limited number).

A one-sample *t*-test was used to test if the difference in the protein levels for baseline samples and the mechanical ventilator were significantly different from zero. 

### 2.8. Bioinformatic Analysis

The biological relationships of differentially expressed proteins were investigated using the Search Tool for the Retrieval of Interacting Genes/Proteins (STRING, string-db.org) for protein–protein interaction enrichment analysis [15]. The gene names as given in Appendix A were entered for recognition as human genes. The protein–protein interaction enrichment *p*-value was calculated based on RGGDS (Random Graph with Given Degree Sequence), as described in [16].

### 2.9. Adjusted Differential Expression Analysis and Prediction Models

In order to expand and confirm the differential expression results obtained using the t-test as described above, a secondary analysis was performed using the statistical software package R v4.0.2 [17]. Protein expression data from MaxQuant were read using the MSnbase framework, and technical replicates were combined by summing data [18,19]. Only proteins present in more than 70% of the samples were included, and remaining missing values were imputed using the k-nearest neighbor approach in MSnbase. Differential expression analysis was performed using the negative binomial model from edgeR [20] implemented in the package msmsTests [21]. *p*-values from the differential expression analysis were adjusted for multiple testing using the Benjamini–Hochberg method to control the false discovery rate. Proteins were considered as being significantly differentially expressed when the adjusted *p*-value was below 0.1. Prediction models using the imputed protein data as features were trained for classifying PE versus non-PE in relevant comparisons. For the multivariate prediction models, the combined, imputed data, as described above, were used. A relative protein expression per individual was obtained by dividing by the total amount of protein in each sample. These values were subsequently log_2_-normalized and centered around the median.

The binary outcomes, i.e., PE animal vs. negative control pigs or Early/Late Post PE versus Pre PE, were fitted using a logistic regression model with an elastic net penalty using the R package glmnet [22]. This penalization enables variable selection and forces the model to make a tradeoff between the residual error in the training set and the number of selected variables, in order to obtain higher predictive power in external datasets. The size of the penalty is controlled using two tuning parameters, where optimal values were obtained using leave-one-out cross-validation with the percentage of misclassified samples in the validation set as the loss function. A receiver operating curve (ROC) was drawn between predicted probabilities of PE from the validation folds in the optimal model from the cross-validation vs. the true labels. The best cut was obtained as the value maximizing the sum of the sensitivity and specificity. The final model for each scenario was fitted using the full dataset and optimal tuning parameters in order to interpret selected variables.

### 2.10. Statistics Concerning Clinical Parameters and Protein Concentrations

Estimates were presented as means and standard error of the mean (SEM). Means were compared using paired or non-paired t-tests as appropriate. A one way-analysis of variance (ANOVA) was used to determine whether the EBC protein concentrations were constant throughout the experiment. The underlying assumptions of normality and variance homogeneity were checked by visual inspection of relevant plots and Bartlett’s test, respectively.

## 3. Results

### 3.1. The Research Animals

Complete EBC samples were collected from 14 PE animals and four negative controls (Figure 2). The mean temperature in the research facility was 22.4 °C. Clinical and para-clinical data for the PE animals and negative controls are summarized in Table 1. The PE animals had hemodynamic signs of right ventricular strain but no signs of acute heart failure, corresponding to intermediate risk PE, while in the negative controls, the hemodynamics were unaltered [2,11].

### 3.2. The Proteins in the EBC

The mean protein concentration in the EBC was 2.83 ± 0.08 µg/mL for the PE animals, with no difference in Pre PE, Early Post PE and Late post PE samples (Appendix A). The mean protein concentration in the EBC from the negative controls was 3.17 ± 0.20 µg/mL, not significantly different from PE animals (Appendix A). The protein concentration in the condensate from the mechanical ventilator was 4.81 µg/mL.

After filtering in Perseus, 897 proteins remained for further analysis. The mean number of identified proteins in the EBCs was 249 (range 53–687). Ninety-one proteins were present in more than 70% of the samples. Of the 897 proteins, 286 (32%) were also identified in the condensate from the mechanical ventilator; no proteins were solely identified in the mechanical ventilator (Appendix A).

### 3.3. Discovery Based Differential Expression Analysis and Bioinformatics

The expression of, in total, 145 proteins changed after the induction of PE compared with baseline samples or negative controls based on two-sided t-tests of the 897 proteins (Appendix A). Fifty of the proteins were present at higher amounts after PE compared with before PE or negative controls with fold changes between 1.3 and 17.1; seven of them in more than one of the comparisons (Appendix A). The bioinformatic analysis of the fifty upregulated proteins using STRING showed significantly more interactions within the group than expected for a random set of 50 proteins from the genome, indicating that the up-regulated proteins are at least partially biologically connected. Most of the proteins were components of cytosol or the cytoplasm (Figure 4).

Ninety-five proteins were present at lower amounts after PE compared with before PE or negative controls with fold changes between 0.1 and 0.8, and 27 of them in more than one of the comparisons (Appendix A). Ninety of the 95 downregulated proteins were recognized in the STRING analysis (EEF1A1P5, IGHG1, IGHG2, IGLC3 and IGKC were not identified). The bioinformatic analysis showed significantly more interactions within the group than expected for 90 randomly selected proteins from the genome, indicating that the down-regulated proteins are at least partly biologically connected. Most of the proteins were components of cytosol or the cytoplasm (Figure 5). 

Twenty-nine proteins differed in the Early Post C and Late Post C samples compared with the Pre C samples, while four proteins differed in both Early Post C and Late Post C compared with baseline C (Appendix A). Thirteen of the 29 proteins were also differentially expressed in samples after PE compared with no PE, but with inverse changes in negative controls for all thirteen proteins (Appendix A).

### 3.4. Sensitivity Analysis

In the Late Post PE samples from PE animals treated with placebo (*n =* 6), 12 proteins were significantly altered compared with Pre PE (Appendix A). Four of these proteins were also identified in one or more of the analyses comparing Early Post PE or Late Post PE with Pre PE in the analysis including all 14 PE animals (LYZ, PSMA6, GSN, LCN2).

### 3.5. Analyses Adjusted for Multiple Comparisons and Prediction Models

The second differential expression analysis adjusted for multiple comparisons, based on the 91 proteins present in at least 70% of all EBC samples, led to the identification of 30 proteins with significantly changed levels after PE (Table 2). Five of the proteins were not identified in the first differential analysis (IGHA1, S100A9, AHNAK, S100A7 and PKM), while the remaining 25 proteins were (Table 2 and Appendix A). None of the 91 proteins in the analysis were significantly differentially expressed in the Late Post PE vs. Late Post C, Early Post C vs. Pre C and Late Post C vs. Pre C comparisons and, consequently, no prediction models were trained for these comparisons.

The optimal prediction models gave cross-validation misclassification errors of 5.5% for Early Post PE vs. Pre C, 3.5% for Early Post PE vs. Pre PE, and 18.5% for Late Post PE vs. Pre PE. The predicted probabilities in the held-out folds from the cross validation gave ROC AUCs of 0.93 and 0.99 and best cut-off values of 0.53 and 0.51 for Early Post PE vs. Early Post C and Early Post PE vs. Pre PE, respectively. For Late Post PE vs. Pre PE, the best cut was found to be at a predicted probability of PE of 0.45 with a sensitivity of 0.85, a specificity of 0.85 and ROC AUC of 0.77 (Figure 6).

Refitting the model to the full data using the optimal parameters for the penalization led to a model with three proteins for Early Post PE vs. Early Post C; ACTG1 and TUBA1B, both with negative coefficients, meaning that an increase in expression decreases the probability of PE, and DSG1, with a positive coefficient, indicating that an increase in DSG1 increases the probability of PE (Appendix A). The model for Early Post PE vs. Pre PE included 67 (Appendix A). Finally, the model for Late Post PE vs. Pre PE had five proteins where high expression of RPLP2, LCN2 and IL36G decreased the probability of PE, while high expression of ALB plus IGHA1 increased the probability of PE (Figure 6, Appendix A).

## 4. Discussion

The results of this study indicate that it is possible to find biomarkers for acute PE in EBC. The PE animals in our study all suffered intermediate-risk PE, while the negative controls had no functional signs of PE. We identified 50 proteins present in the EBC at higher amounts in PE and 95 proteins present at lower amounts in PE, 25 of these proteins remained differentially expressed in analysis restricted to proteins present in at least 70% of samples, with imputation and adjusted for multiple comparisons. This analysis added another four proteins to the total list of putative exhaled markers for PE.

The final prediction model for distinguishing Pre PE from Late Post PE was based on only five proteins and was able to classify 22 out of 27 EBC samples correctly in the cross validation with a sensitivity of 0.85 and a specificity of 0.79. Since parameters in the cross-validation have been optimized to correctly classify the held-out samples, this accuracy is probably overestimated, and an external dataset is needed for confirmation of the results.

Bioinformatic analysis indicated that both the upregulated and downregulated proteins were at least partially related and showed that most of the proteins were part of intracellular compartments. 

This is the first study to investigate the potential of proteins in the EBC as diagnostic biomarkers for PE, but physicians have known for centuries that the exhaled breath can contribute to information in diseases. In 1798, John Rollo described the “odor of decaying apples” in patients with diabetes mellitus [23]. Several breath-analysis tests have been developed and approved by the FDA, but they mainly focus on volatile organic compounds, as did a proof-of-principle study on pulmonary embolism [24,25]. 

The proteome of the EBCs collected in our porcine model of intermediate-risk PE was overall very consistent with descriptions of the proteome of EBC from pooled samples from humans: we identified 32% (14 of 44), 80% (116 of 145) and 76% (127 of 167), respectively, of identified proteins in previous studies [8,10,26]. The estimated protein concentration in the EBC is comparable to other reports, and to the initial findings in our pilot study [9,27,28,29].

Hypoxia and direct damage to the lung tissue due to pulmonary embolism causes cell death/destruction [30]. Intracellular contents are released in such circumstances, mainly from cytoplasm [31]. This was in accordance with our findings showing that intracellular proteins were especially upregulated after PE. Furthermore, we detected the upregulation of many major plasma proteins and hemoglobin after PE (e.g., ALB, globulins, HPX, TF, HBB, HBA). We speculate that this reflects the ischemic pulmonary parenchymal necrosis, which leads to increased diffusion across the blood–air barrier. This will lead primarily to an abundance of plasma proteins in the EBC, in fact a kind of “micro hemoptysis”, but rarely to macroscopic hemoptysis, which is considered to be a clinically relevant symptom of PE.

Most of the putative markers in our study were present at lower amounts after PE. These proteins can be useful in future diagnostics of PE in a calculated diagnostic ratio of downregulated protein to upregulated protein in the EBC, such as, for example, the urine albumin-to-creatinine ratio or the light chain ratio in blood. 

Several of the putative markers in this study have been described in previous studies in different aspects of thrombosis. Bleomycin hydrolase (BLMH) was recently identified as a novel diagnostic plasma biomarker for deep vein thrombosis [32]. In untargeted proteomic analysis of blood samples, Insenser et al. found higher levels of IGHA1 in patients with low-risk PE compared with high-risk PE [33]. The levels of APOA1 and ALB were significantly higher in patients with deep vein thrombosis than in healthy controls in un-targeted protein analysis of urine samples [34]. Nearly half of the upregulated proteins in our study (CLU, TF, HPX, HBB, ALB, FN1, LYZ, PSMA3, PITHD1, APOA1, CDC42, G6PD, ACTB, VCL, HNRNPK, CAPN1, AZGP1, TGM1, EEF1G, PRDX1, KRT2 and HBA) were identified in studies of the composition of thrombus material [35,36]. One of the most downregulated proteins in our study was LCN2. This protein is found to be upregulated in plasma samples from patients with COPD and in patients with myocardial dysfunction [37,38]. However, this study was not designed to identify specific markers of PE, and some of the putative markers may unspecifically indicate general pulmonary damage or cardiovascular strain. 

### 4.1. Study Limitations

The lack of standardization is the most important limitation of our study [39], and it is thus mainly an initial proof-of-concept for the potential of diagnostic biomarkers of pulmonary embolism in the EBC. Other limitations must also be considered. First, when a protein is not identified in the MS, it is not necessarily because it is not present in the sample because of limited sensitivity of the method. Second, the vasodilatory drugs administered to eight of the pigs before the collection of the Late Post PE sample might have influenced the protein composition in the EBCs from these animals. In the sensitivity analysis based on EBC from placebo animals, we did, however, find expression of many of the same proteins as in the full dataset. Third, we cannot rule out that some of the proteins being higher in the Pre PE as compared with the Early Post PE and Late Post PE samples (i.e., lower after PE) was due to “left overs” from earlier pigs being mechanically ventilated on the same machine. However, only 32% of the total number of identified proteins in the EBC from the pigs were identified in the condensate from the mechanical ventilator, and no proteins were solely identified in the mechanical ventilator. Furthermore, this problem should be minimal, since the pigs were mechanically ventilated for more than three hours on average before the Pre PE samples were collected.

We did not measure biochemical markers of pulmonary embolism diagnosis/prognosis in the blood of the pigs in this explorative study (i.e., D-dimer, troponins), this will be included in future clinical studies.

### 4.2. Perspectives

A diagnostic test for PE based on EBC analysis has the potential to improve early diagnosis and treatment in acute PE. Furthermore, a diagnostic test based on EBC analysis would dramatically improve the diagnostic process for those patients in whom CT-scans are not an option. Optimal diagnostic markers of PE would enable the diagnosis of PE in patients with symptoms shared with other acute cardiothoracic and respiratory diseases, possibly already in the pre-hospital setting. Our idea of diagnosing PE in the EBC thus has a large potential for improving the diagnosis of PE in many settings, even in the pre-hospital setting. Low cost, easy access and use are key features for a new diagnostic test to perfectly complement the existing methods. A point-of-care test (POCT) based on proteins in the exhaled breath could provide a valuable extension of the diagnostic methods for pulmonary embolism, and possibly also other pulmonary diseases.

Analysis of the exhaled breath is already used in other diseases for diagnosis or monitoring. This is the first study ever to investigate the protein content of the EBC in PE, and our data suggest that analysis of the EBC has the potential to be a new method for the non-invasive diagnosis of PE based on the calculation of a ratio of proteins. Studies based on human EBC samples are needed to determine whether this translates clinically.

## 5. Conclusions

In conclusion, our study indicates that the EBC holds several putative biomarkers for acute intermediate risk PE.

## 6. Patents

The novelty and innovative level of the findings presented in our study resulted in a recently published patent application: (https://patentscope.wipo.int/search/en/detail.jsf?docId=WO2020245200&tab=PCTBIBLIO, accessed on 28 October 2021). The patent applicants are Aalborg University Hospital, Aarhus University and Aalborg University. The inventors are Inger Lise Gade, Søren Risom Kristensen and Bent Honoré. 

## Figures and Tables

**Figure 1 jcm-10-05165-f001:**
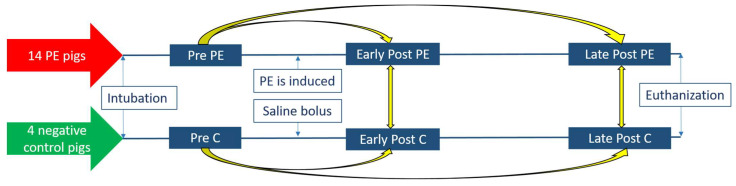
Timing of the collection of the exhaled breath condensate samples and overview of the comparisons (yellow). Abbreviations: PE = pulmonary embolism, C = negative control.

**Figure 2 jcm-10-05165-f002:**
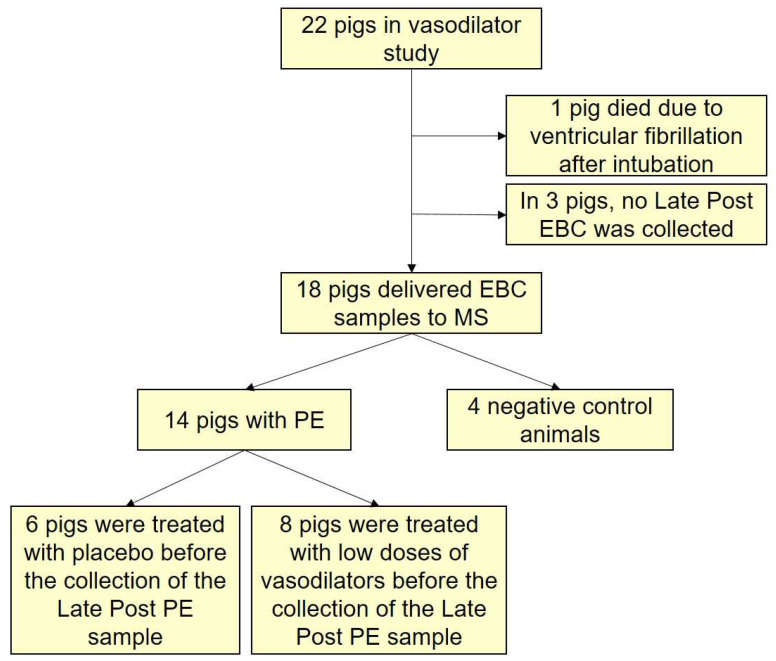
Overview of the study animals. Abbreviations: PE = pulmonary embolism, MS = mass spectrometry, EBC = exhaled breath condensate.

**Figure 3 jcm-10-05165-f003:**
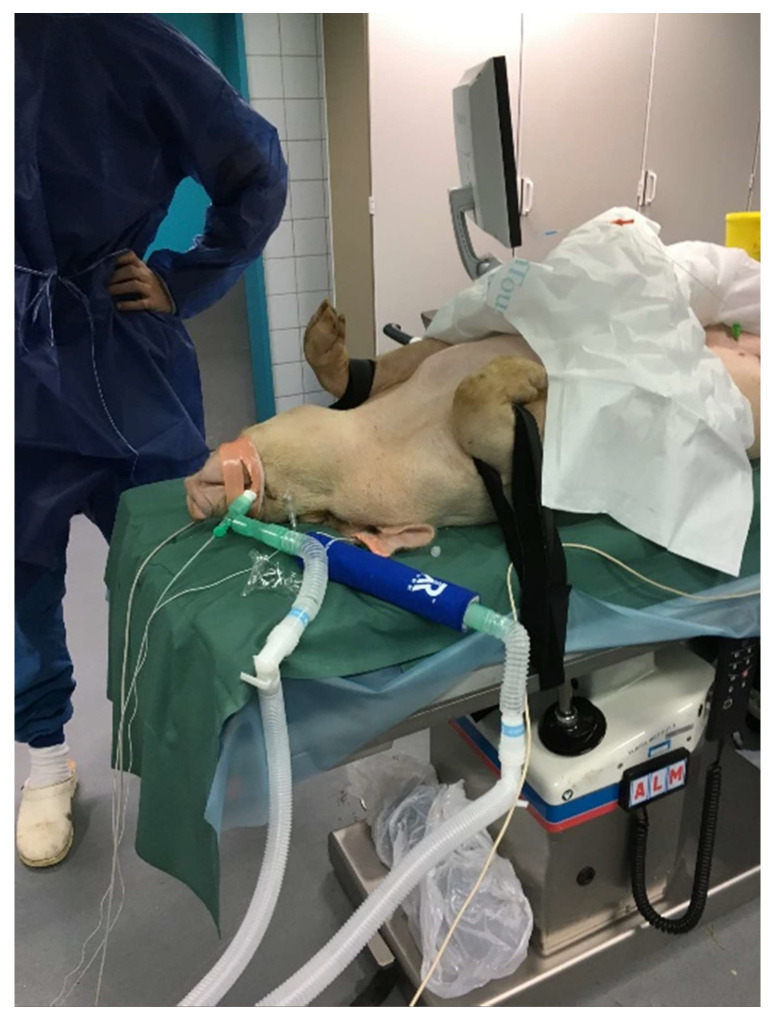
The Rtube was inserted between the expiratory limb of the Y-connector and the expiratory limb of the mechanical ventilator.

**Figure 4 jcm-10-05165-f004:**
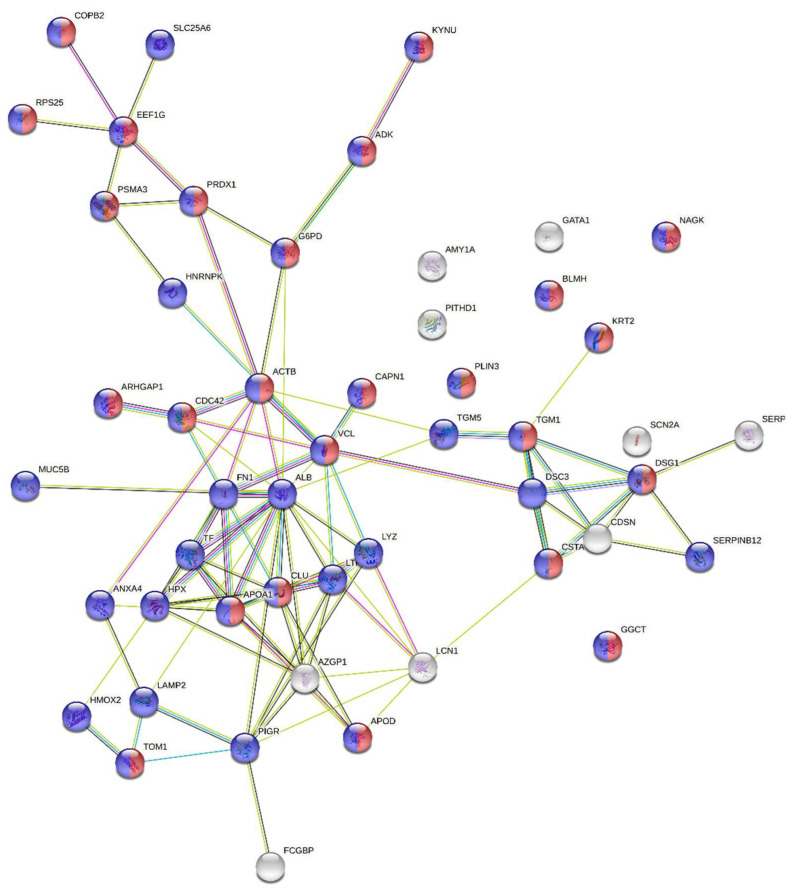
Bioinformatic characterization of proteins present in the EBC at higher amounts after PE (*n =* 50). The number of edges is 98. For a random set of 50 proteins, 34 edges would be expected. The number of interactions is significantly higher than expected (protein–protein interaction enrichment *p*-value < 1 × 10^−16^). Blue nodes: cytoplasm proteins (41). Red nodes: cytosol proteins (25). The protein–protein interaction enrichment *p*-value was estimated based on a Random Graph with Given Degree Sequence (RGGDS), as described in [16].

**Figure 5 jcm-10-05165-f005:**
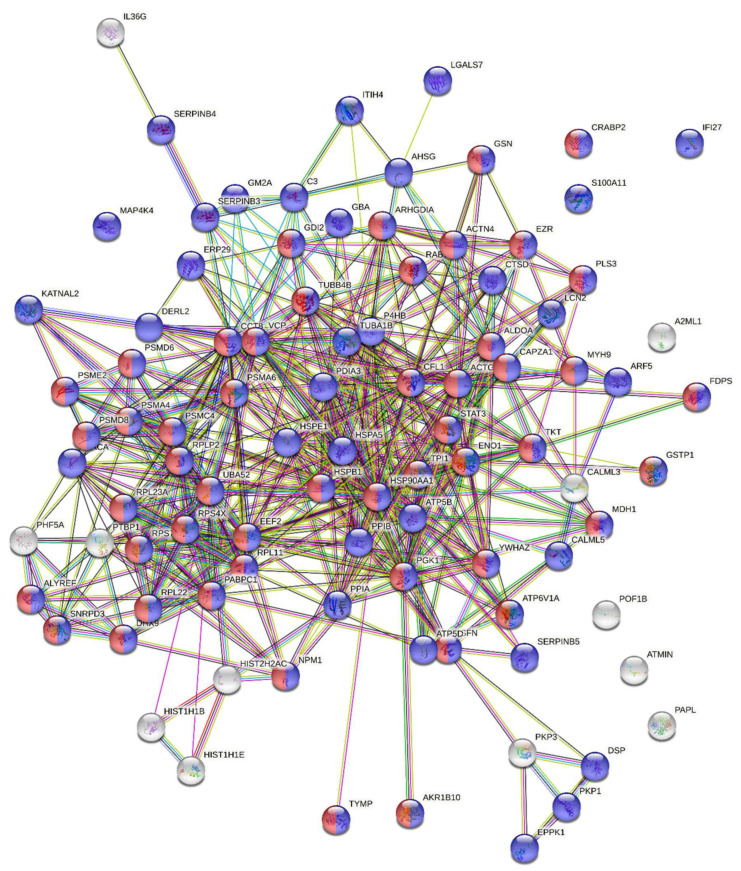
Bioinformatic analysis of proteins present in the EBC at lower amounts in PE (*n =* 90). The number of edges is 500. For a random set of 90 proteins, 195 edges would be expected. The number of interactions is significantly higher than expected (*p* < 1 × 10^−16^). Blue nodes: cytoplasm proteins (82). Red nodes: cytosol proteins (50). The protein–protein interaction enrichment *p*-value was estimated based on a Random Graph with Given Degree Sequence (RGGDS), as described in [16].

**Figure 6 jcm-10-05165-f006:**
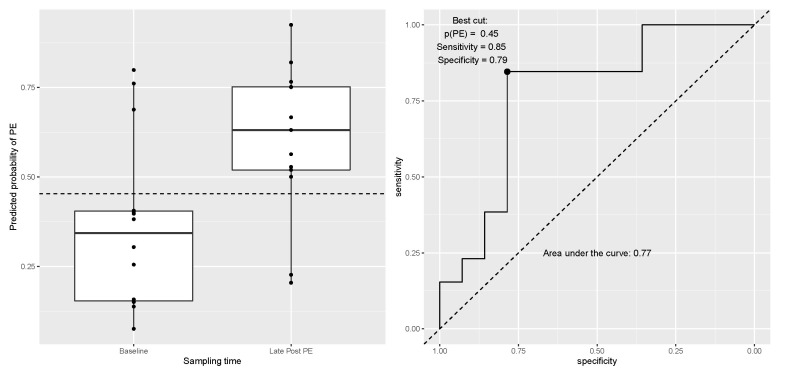
Boxplot and ROC curve for predicted probability of PE vs. true status at late post for PE samples. Predicted probabilities were obtained from the hold-out folds for the optimal model in the cross validation. Abbreviations: ROC = receiver operating characteristic, PE = pulmonary embolism, MS = mass spectrometry.

**Table 1 jcm-10-05165-t001:** Clinical data, functional and invasive measures at baseline and 30 min after pulmonary embolism for PE animals and at index times for the negative controls. (mean ± SEM).

	PE Animals (*n =* 14)			Negative Controls (*n =* 4)		
	Before PE (Baseline)	Early Post PE	*p*-Value	Baseline C	Early Post C	*p*-Value
Heart rate (beats/min)	62.3 ± 2.9	70.9 ± 5.0	0.09	59.3±5.4	57.5±5.2	0.57
Pulmonary vascular resistance (dyn·s/cm^5^)	99.1 ± 4.6	332.8 ± 18.1	<0.01	139.7 ± 36.1	164.4 ± 42.8	0.06
Mean pulmonary arterial pressure (mmHg)	18.3 ± 0.5	35.0 ± 1.1	<0.01	20.0 ± 2.4	21.8 ± 1.5	0.19
Mean Arterial blood pressure (mmHg)	82.9 ± 3.0	84.3 ± 4.0	0.49	79.8 ± 7.4	75.8 ± 5.5	0.32
Cardiac Output (L/min)	5.67 ± 0.22	6.26 ± 0.24	<0.01	5.23 ± 0.12	5.48 ± 0.13	0.37
RV/LV end systolic	0.64 ± 0.04	1.29 ± 0.05	<0.01	0.63 ± 0.03	0.61 ± 0.02	0.73
End tidal-CO_2_ (kPa)	5.1 ± 0.1	4.9 ± 0.1	0.06	5.0 ± 0.1	5.0 ± 0.3	0.77
pH	7.50 ± 0.01	7.23 ± 0.21	0.23	7.54 ± 0.01	7.55 ± 0.02	0.65
PaCO_2_ (kPa)	4.58 ± 0.07	5.40 ± 0.17	<0.01	4.55 ± 0.15	4.68 ± 0.12	0.57
PaO_2_ (kPa)	16.60 ± 0.80	12.82 ± 0.84	<0.01	18.05 ± 0.32	17.05 ± 0.34	0.06
Rectal temperature	38.7 ± 0.1	39.0 ± 0.2	0.08	39.3 ± 0.5	39.2 ± 0.3	0.8
Weight (kg)	57.3 ± 0.9	-	-	56.9 ± 0.4	-	-

Abbreviations: PaCO_2_ = arterial CO_2_ tension, PaO_2_ = arterial O_2_ tension, RV/LV = right ventricle end systolic volume divided by left ventricle end systolic volume, SEM = standard error of the mean, PE = pulmonary embolism, C = negative control.

**Table 2 jcm-10-05165-t002:** Differential expression analysis based on 91 proteins with imputation of missing values and adjusted for multiple comparisons.

Majority Protein IDs	Protein Name	Gene Name	Early Post PE vs. Pre PE		Late Post PE vs. Pre PE		Early Post PE vs. Early Post C	
Positive Fold Changes (i.e., Higher Amount after PE)			Log Fold Change	*p*-Value	Log Fold Change	*p*-Value	Log Fold Change	*p*-Value
P01876	Immunoglobulin heavy constant alpha 1	IGHA1	4.58	<0.001	4.03	<0.001		
P31025	Lipocalin-1	LCN1	2.51	0.005				
P61626	Lysozyme C	LYZ			3.29	0.019		
Negative fold changes (ie lower amount after PE)								
P08835	Albumin	ALB					−4.54	0.0134
P55072	Transitional endoplasmic reticulum ATPase	VCP	−1.82	0.025				
P19971	Thymidine phosphorylase	TYMP	−2.76	0.003				
Q2XVP4	Tubulin alpha-1B chain	TUBA1B	−2.71	0.003			−2.70	0.007
A8K2U0	Alpha-2-macroglobulin-like protein 1	A2ML1	−2.05	0.003				
P58107	Epiplakin	EPPK1	−2.59	0.003				
P48594	Serpin B4	SERPINB4	−1.90	0.003				
P68371	Tubulin beta-4B chain	TUBB4B	−1.98	0.013				
P35579	Myosin-9	MYH9	−2.34	0.002				
Q5VTE0	Putative elongation factor 1-alpha-like 3	EEF1A1P5	−1.85	0.019				
P13639	Elongation factor 2	EEF2	−2.05	0.010				
P01857	Ig gamma-1 chain C region	IGHG1	−2.06	0.006	−2.30	0.019		
P47929	Galectin-7	LGALS7	−1.98	0.004				
P06733	Alpha-enolase	ENO1	−1.22	0.032				
P60174	Triosephosphate isomerase	TPI1	−1.81	0.002				
P00558	Phosphoglycerate kinase 1	PGK1	−1.70	0.016				
P05387	60S acidic ribosomal protein P2	RPLP2	−1.24	0.019				
P06702	Protein S100-A9	S100A9	−2.16	0.032				
P14618	Pyruvat kinase	PKM	−1.67	0.032				
P80188	Neutrophil gelatinase-associated lipocalin	LCN2	−1.33	0.035	−2.10	0.003		
P06576	ATP synthase subunit beta, mitochondrial	ATP5B	−1.78	0.035				
P01024	Complement C3	C3	−1.27	0.048				
P09211	Glutathione S-transferase P	GSTP1	−1.19	0.062				
Q09666	Neuroblast differentiation-associated protein AHNAK	AHNAK	−1.59	0.062				
P29508	Serpin B3	SERPINB3	−1.15	0.063				
P31151	Protein S100-A7	S100A7	−1.16	0.077				
O02705	Heat shock protein HSP 90-alpha	HSP90AA1	−1.20	0.085				

## Data Availability

The data presented in this study will openly available in (repository) after publication of this paper.

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
