# Peer review of "Putative Biomarkers for Acute Pulmonary Embolism in Exhaled Breath Condensate"

_jcm, 2021, doi:10.3390/jcm10215165_

Round 1

Reviewer 1 Report

  1. Lines 54-58: you mention that you collected two EBC samples from each pig in every step of the study. Did you mean the exhaled sample and the ventilator sample? Or two exhaled samples and then you calculated the mean value of the exhaled proteins?
  2. Lines 61-63: you implemented vasodilatatory drugs before the collection of the Late Post PE sample. No comparison was made before and after the vasodilatatory effect. So please explain the necessity for this pharmaceutical intervention.
  3. Figure 1: explain the reason of euthanization, especially in the control group.
  4. Lines 96-98: no positive end-expiratory pressure was implemented during the invasive mechanical ventilation. Is there any pathophysiological background for this decision?
  5. Figure 2: Please correct the word vasodilaltor in the top of the flow chart.
  6. Figure 2: Please try to keep the same nomenclature in the bottom of the flow chart referring to: Late Post PE ECB sample vs. Late post PE sample.
  7. Lines 180-182: you accepted proteins identified with at least one peptide. What do you mean exactly?
  8. Lines 248-250: you decided to induce an intermediate risk PE with right ventricular strain but not a right ventricular insufficiency. Why did you avoid to work with high risk PE? Somebody would expect that your results could be stronger. Did you believe that some of the proteins are influenced if a right ventricular insufficiency is present?

Author Response

Reply to Reviewer 1

First and foremost, thanks for a thorough and insightful review of our work. Your effort is much appreciated!

  1. Lines 54-58: you mention that you collected two EBC samples from each pig in every step of the study. Did you mean the exhaled sample and the ventilator sample? Or two exhaled samples and then you calculated the mean value of the exhaled proteins?

RE: We collected two exhaled samples at each step for each pig in order to have as much protein as possible for the analysis. The two EBC samples were pooled into 2 mL cryo-tubes and stored at -80 ºC until analyzed as one sample with regards to protein concentration and in the subsequent mass spectrometry analysis. The samples were run as replicates in the mass spectrometry analysis (one repeated measurement of the same sample). In the analysis of the protein data, the LFQ values from the two measurements were averaged by the mean, except for five of the samples. This has been elaborated in the manuscript: Lines 57-59 The two EBC samples from each time point were pooled into 2 mL tubes and stored at -80 ºC until analyzed as one sample with regards to protein concentration and in the subsequent mass spectrometry analysis.”

  1. Lines 61-63: you implemented vasodilatatory drugs before the collection of the Late Post PE sample. No comparison was made before and after the vasodilatatory effect. So please explain the necessity for this pharmaceutical intervention.

RE: We aimed at investigating the potential for diagnosing PE based on protein analyses of the exhaled breath condensate. Our study is an amendment protocol to a pharmaco-kinetic study of three different vasodilatory drugs (inhaled nitric oxide, Sildenafil and Riociguate) in a porcine model of pulmonary embolism. In total four increasing doses of vasodilatory drugs (clinically equivalent) were given in the pharmacological study. We collected EBC samples from the negative control pigs (no PE, n=4), the pigs treated with placebo (isotonic saline, n=6) and immediately after administration of the second (still low) dose of vasodilatory drug (n=8). No EBC samples were collected after the third and fourth dose of vasodilatory drugs.

Since investigations of vasodilatory drugs was not the scope of this study, we did not include comparisons of EBC before and after vasodilators in our manuscript. We did however include analysis restricted to the PE animals treated with placebo in order to see if the observed differences were explained solely because of administration of (ultra-low) doses of vasodilators before the collection of Late Pose PE samples (section 3.4. Sensitivity analysis in the manuscript). We rephrased the 2.1. Study design section in order to make this clearer.

  1. Figure 1: explain the reason of euthanization, especially in the control group.

RE: All the pigs, including the negative controls, had 26 French catheters inserted directly into the central veins. This is a big cathether, and the pigs would have a very high risk of severe internal bleeding in case we extubated the pigs and removed the catheter. It was therefore decided to euthanize all the animals.

  1. Lines 96-98: no positive end-expiratory pressure was implemented during the invasive mechanical ventilation. Is there any pathophysiological background for this decision?

RE: The pigs underwent invasive measurements during the experiment, and we wanted these measurements to be as precise a s possible. We did not implement positive end expiratory pressure in order to avoid “false high” intra-thoracical pressure, which would increase the pulmonary vascular resistance (further). The pigs were mechanically ventilated for a maximum of four hours, and we monitored the expiratory CO2 levels continuously but cannot rule out that the pigs had small pulmonary atelectasis. We consider this of minor importance compared to the massive pulmonary emboli we induced in the pigs.

  1. Figure 2: Please correct the word vasodilaltor in the top of the flow chart.

RE: Thank you, it has been corrected.

  1. Figure 2: Please try to keep the same nomenclature in the bottom of the flow chart referring to: Late Post PE ECB sample vs. Late post PE sample.

RE: Thanks again, the figure text has been revised.

  1. Lines 180-182: you accepted proteins identified with at least one peptide. What do you mean exactly?

RE: Protein data can be filtered based on how many specific peptides one observe in a sample. We were interested in all possible proteins in the exhaled breath condensates in this exploratory study, and proteins identified based on (only) one specific peptide were therefore included. This has been elaborated in the manuscript, lines 188-190: “(i.e. we did not remove proteins if they were not found in all the EBC samples and we accepted proteins identified with one peptide because we were interested in all possible proteins in the samples in this explorative study).”

  1. Lines 248-250: you decided to induce an intermediate risk PE with right ventricular strain but not a right ventricular insufficiency. Why did you avoid to work with high risk PE? Somebody would expect that your results could be stronger. Did you believe that some of the proteins are influenced if a right ventricular insufficiency is present?

RE: The pigs had to have pulmonary emboli big enough to make them quite sick, but we had to balance it so they did not die before the experiment was finished. The entire protocol took about 2 hours(Schultz, J.; Andersen, A.; Gade, I.L.; Kjaergaard, B.; Nielsen-Kudsk, J.E. Riociguat, sildenafil and inhaled nitric oxide reduces pulmonary vascular resistance and improves right ventricular function in a porcine model of acute pulmonary embolism. Eur. Hear. J. Acute Cardiovasc. Care 2019, doi:10.1177/2048872619840772.) We tried several “doses” of pulmonary emboli in pilot studies, and learned that two autologous emboli would affect the pigs corresponding to intermediate-high-risk pulmonary emboli. If we gave three emboli, a considerable proportion of the experimental pigs would die before the experimental protocol was finished.

We think your point about possibly stronger results in a high-risk model is interesting and probably correct, but for both animal-ethics and economic reasons this was not an option for us.

Our best guess is, that we would see even higher levels of common blood proteins (e.g. albumin, hemoglobins and haptoglobin) in the EBC samples in case of right ventricular insufficiency.

Reviewer 2 Report

The authors presented interesting results. The manuscript is well written and give an imporntante message to translate into clinicla practice.

I have only some suggestions:

  • Did the auhtors evalauted at base-line the mean troponin value? As the auhtors know it is an imporntant and foundamental marker for PE prognostic stratification; otherwise it would be stated into the limitations
  • Are other TTE values avaialable (i.e., TAPSE, IVC diameter etc..)?
  • I sugggest to more deeply discuss the potential use of such diagnostic approach in future clinical practice nd relative limitations (costs, logistic and material requirements, learning curve, etc)

Author Response

Reply to Reviewer 2

The authors presented interesting results. The manuscript is well written and give an imporntante message to translate into clinicla practice.

RE: Thank you for the positive comments, and for your time and effort.

I have only some suggestions:

  • Did the auhtors evalauted at base-line the mean troponin value? As the auhtors know it is an imporntant and foundamental marker for PE prognostic stratification; otherwise it would be stated into the limitations

RE: We measured the troponin levels when we developed the porcine model of pulmonary embolism, where we saw increased troponin-T levels after induction of the PE: https://www.ncbi.nlm.nih.gov/pmc/articles/PMC5798692/pdf/10.1177_2045893217738217.pdf ) In this study, however, we used the TTE and invasive hemodynamic measures of right ventricular strain, but we must of course include troponin measures in our clinical study. I have included this in the discussion lines 417-419: We did not measure biochemical markers of pulmonary embolism diagnosis/prognosis in the blood of the pigs in this explorative study (i.e. d-dimer, troponins), this will be included in future clinical studies.

  • Are other TTE values avaialable (i.e., TAPSE, IVC diameter etc..)?

RE: We did not perform TTE, only transoesophageal echocardiography (TEE) was done in our study. The procedure was a challenge due to the longitudinal axis of the porcine heart. The only parameter we obtained from the TEE was therefore the right ventricle end systolic volume divided by left ventricle end systolic volume at the end-diastole. We have made this clear in the manuscript. Line 108-110: “The right ventricle end systolic volumes and the left ventricle end systolic volumes were measured by transoesophageal echocardiography.”

  • I sugggest to more deeply discuss the potential use of such diagnostic approach in future clinical practice nd relative limitations (costs, logistic and material requirements, learning curve, etc)

RE: Thank you for this suggestion and opportunity. We have included our thoughts on this in the discussion, lines 426-431: “Our idea of diagnosing PE in the EBC thus has a large potential for improving the diagno-sis of PE in many settings, even in the pre-hospital setting. Low cost, easy access and - use are key features for a new diagnostic test to perfectly complement the existing methods. A Point-of-Care Test (POCT) based on proteins in the exhaled breath could provide a valuable extension of the diagnostic methods for pulmonary embolism, and possibly also other pulmonary diseases.”

Round 2

Reviewer 1 Report

Dear Authors,

thank you for providing comprehensive and convincing answers to my questions and queries. You rephrased and/or elucidated important key-points of your study and revised the manuscript, which has contributed to quality improvement and increased the publishing potential of your manuscript.

Best Regards